# A Combination of Three Genomic Regions Conditions High Level of Adult Plant Stripe Rust Resistance in Australian Wheat Cultivar Sentinel

**DOI:** 10.3390/plants13010129

**Published:** 2024-01-02

**Authors:** Bosco Chemayek, William Wagoire, Urmil Bansal, Harbans Bariana

**Affiliations:** 1Plant Breeding Institute, School of Life and Environmental Sciences, Faculty of Science, The University of Sydney, 107 Cobbitty Road, Cobbitty, NSW 2570, Australia; bchemayek@gmail.com (B.C.); urmil.bansal@sydney.edu.au (U.B.); 2Buginyanya Zonal Agricultural Research Institute, National Agricultural Research Organisation, Mbale P.O. Box 1356, Uganda; wwwagoire@gmail.com; 3School of Science, Hawkesbury Campus, Western Sydney University, Bourke Street, Richmond, NSW 2753, Australia

**Keywords:** APR, QTL mapping, QTL interaction, stripe rust, wheat

## Abstract

A seedling susceptible Australian common wheat cultivar Sentinel showed resistance to stripe rust under field conditions. A Sentinel/Nyabing3 (Nyb3)-derived recombinant inbred line (RIL) population was phenotyped. A DArTseq marker-based linkage map of the Sentinel/Nyb3 RIL population was used to determine the chromosomal location of the adult plant stripe rust resistance possessed by Sentinel. Three consistent quantitative trait loci (QTL); *QYr.sun-1BL*, *QYr.sun-2AS* and *QYr.sun-3BS* were detected, and they on an average explained 18%, 15.6% and 10.6% of the variation in stripe rust response, respectively. All three QTL were contributed by Sentinel. *QYr.sun-1B* corresponded to the previously characterized gene *Yr29*. Sentinel expressed resistance at the four-leaf stage at 21 ± 2 °C in the greenhouse. Monogenic segregation among the RIL population was observed when screened at the four-leaf stage at 21 ± 2 °C in the greenhouse, and the underlying resistance locus was temporarily named *YrSen*. *QYr.sun-3BS* peaked on *YrSen*. *QYr.sun-2AS* was mendelized by generating and phenotyping a mongenically sgregating F_6_ RIL population, and it was temporarily designated *YrSen2*. RILs carrying *Yr29*, *YrSen* and *YrSen2* in combination exhibited responses like the parent Sentinel. Based on a comparison of the genomic locations and resistance expression with stripe rust resistance genes previously located in their respective chromosomes, *QYr.sun-2AS* (*YrSen2*) and *QYr.sun-3BS* (*YrSen*) were concluded to represent new loci.

## 1. Introduction

Stripe rust of wheat, caused by *Puccinia striiformis* f. sp. *tritici* (*Pst*), is one of the most devastating diseases of wheat and threatens the global food supply [1]. New pathogen genotypes are more aggressive and can infect previously resistant wheat varieties, leading to rapid pathogen migration across the continents [2]. *Pst* infects leaf tissue and significantly reduces grain yield and quality in susceptible cultivars. More than 88% of the global wheat cultivars are now susceptible to stripe rust, and yield losses are severe if infection occurs at the early stages of growth [3]. Stripe rust was first detected in eastern Australia in 1979 [4]. The wheat growing areas of Western Australia remained free from stripe rust for more than two decades, but a new exotic pathotype, 134 E16A+, with virulence for *Yr6*, *Yr7*, *Yr8*, *Yr9* and *YrA* and avirulence for *Yr3* and *Yr4*, was detected in 2002 [5]. This pathotype spread to all wheat growing regions in Australia in 2003 and caused epidemics in eastern Australia [6]. The pathotype 134 E16A+ acquired virulence for stripe rust resistance genes *Yr10*, *Yr17*, *Yr24*, *Yr27*, *YrJ* and *YrT* [6].

The new pathotypes of *Pst* have also caused significant wheat yield losses in most parts of the world in recent years and resulted in global losses estimated to be at least 5.5 million tons per year [3]. In 2009–2010, the outbreak of a pathotype of *Pst* with virulence for *Yr27* caused significant yield losses in Azerbaijan, Ethiopia, Iran, Iraq, Kenya, Morocco, Syria, Turkey and Uzbekistan, threatening the food security and livelihood of resource-poor farmers [7,8]. Consequently, there has been significant use of chemicals to control stripe rust epidemics [9]. Genetic resistance is the most economical and environmentally safe means of stripe rust control; hence, the deployment of combinations of stripe rust resistance genes in new wheat cultivars is the best strategy to mitigate potential yield losses [10,11].

Resistance to stripe rust is categorized into two groups: qualitative/all-stage resistance (ASR) and quantitative/adult plant resistance (APR) [12,13]. The ASR is controlled by genes with major effects, and a single gene can provide a high level of protection against avirulent pathotypes of the target pathogen. In contrast, minor effect genes condition APR, and combinations of two or more genes are needed to achieve a commercially acceptable level of resistance [14,15]. Of the more than 80 stripe rust resistance genes formally named so far, only a few represent the APR category [https://wheat.pw.usda.gov/GG3/wgc; 25 October 2023]. The ease of transfer of ASR genes into commercial cultivars has made them a popular choice among wheat breeders; however, new pathotypes often overcome this type of resistance in due course. To achieve long-lasting control of stripe rust, the deployment of combinations of ASR and APR genes in future wheat cultivars is essential. This highlights the need to identify, characterize and map more APR genes.

Cytogenetic techniques were successfully employed to determine the genomic locations of many ASR genes for stripe rust in wheat [16]; however, their role in determining the chromosomal locations of APR genes has been limited due to the poor expression of APR genes in the heterozygous state. Recent advances in high-throughput (next-generation) sequencing platforms such as genotyping by sequencing (GBS) have led to affordable options for whole-genome sequencing using a large number of markers [17,18]. The rapid technological advances in sequencing facilitated the detection of new QTL for stripe rust resistance in common wheat and durum wheat [19,20].

The Australian common wheat cultivar Sentinel has remained resistant to stripe rust under field conditions since its release in 2005, and the genetic basis of this resistance was unknown. This study focused on molecular mapping of APR to stripe rust in Sentinel.

## 2. Results

### 2.1. Stripe Rust Response Assessments

#### 2.1.1. Field Assessment

The resistant parent Sentinel produced adult plant stripe rust response ‘2’, whereas the susceptible parent Nyabing 3 (Nyb3) was scored ‘9’ in the field, when tested with *Pst* pathotype 134 E16A+17+27+ (Figure 1a). The adult plant stripe rust responses of Sentinel/Nyb3 RILs varied from ‘2’ to ‘9’ in all experiments and showed a normal distribution with both parents falling at the tail ends of the curve (Figure 2). Stripe rust response data of the Sentinel/Nyb3 RIL population were subjected to estimation of the number of resistance loci involved using the modified Wright’s method. The involvement of two to three loci in controlling stripe rust response variation among Sentinel/Nyb3 RILs was estimated.

#### 2.1.2. Greenhouse Assessment

Sentinel expressed high level of resistance in the field, and to discount the possibility of involvement of any ASR gene(s), Sentinel and Nyb3 were tested at the two-leaf stage in the greenhouse. Both cultivars produced high infection type (IT) 3+. These results demonstrated the absence of seedling resistance in both genotypes. Both parents were then tested at the two-leaf, three-leaf and four-leaf stages, and post-inoculation incubations were performed under two temperature regimes (17 ± 2 °C and 21 ± 2 °C). Both Sentinel and Nyb3 produced IT3+ at the post-inoculation temperature 17 ± 2 °C at all growth stages. In the 21 ± 2 °C post-inoculation experiment, Nyb3 produced susceptible IT3+ at all growth stages. In contrast, Sentinel exhibited susceptible responses (IT3+) at the two-leaf and three-leaf growth stages, and it displayed IT23C at the four-leaf stage (Figure 1b). The entire RIL population was tested at the four-leaf stage with 21 ± 2 °C as the post-inoculation temperature, and it showed monogenic segregation (58 homozygous resistant (HR; IT23C): 59 homozygous susceptible (HS; IT3+), χ^2^_1:1_ = 0.01, non-significant at *p* = 0.05 and 1 *d.f*.) (Table 1). The resistance locus was tentatively named *YrSen*.

### 2.2. Construction of Linakge Map and QTL Analysis

#### 2.2.1. Detection of APR Genes *Yr18* and *Yr29*

APR genes *Yr18* and *Yr29* are widespread in wheat germplasm. To check whether Sentinel carries any of these genes, linked markers were used. The *Yr18*-linked STS marker *csLV34* amplified a 150 bp PCR product in the positive control Janz and a 229 bp product in Sentinel and Nyb3. Six *Yr18* gene-specific markers (*ccsfr1-cssfr6*) were also tested on Sentinel and Nyb3 DNA to further confirm the absence of *Yr18* in Sentinel. The dominant marker *cssfr1* resulted in amplification of a 517 bp amplicon in Janz and no amplicon in Sentinel and Nyb3, while *cssfr2* amplified a 523 bp amplicon in Sentinel and Nyb3, and no amplicon was recorded in Janz. The co-dominant marker *cssfr3* produced two bands (150 bp and 517 bp) in Janz and a single band (229 bp) in Sentinel and Nyb3. Another such marker, *cssfr4*, amplified a 150 bp PCR product in Janz and two amplicons (229 bp and 523 bp) in Sentinel and Nyb3. The third co-dominant marker, *cssfr5*, resulted in amplification of a 751 bp product in Janz and a 523 bp amplicon in both Sentinel and Nyb3. CAPS marker *cssfr6* produced three amplicons (63 bp/135 bp/451 bp) in positive control Janz and two (63 bp and 589 bp) in Sentinel and Nyb3, following digestion with the restriction enzyme *Fnu4HI*. These results demonstrated the absence of *Yr18* in both Sentinel and Nyb3. The *Yr29*-linked SNP marker *Lr46_SNP1G22* amplified allele ‘A’ in Sentinel and the positive control Lalbahadur + *Lr46*, whereas the alternate allele ‘G’ was detected in Nyb3 and the negative control Lalbahadur. The entire RIL population was tested with *Lr46_SNP1G22*, and monogenic segregation (68 ‘A’: 49 ‘G’; χ^2^
_1:1_ = 3.09, non-significant at *p* = 0.05 and 1 *d.f*.) was noted.

#### 2.2.2. Linkage Map Construction

The RIL population was subjected to whole-genome profiling with 16,815 DArTseq markers. A total of 4891 DArTseq markers exhibited segregation distortion, 1502 had more than 10% missing data, and 2079 were monomorphic between parental genotypes and were therefore not included in the linkage map. Of the remaining 8343 markers, chromosome locations were known only for 3605, whereas 4738 markers were not assigned to any chromosomes and were labeled UK (unknown). Redundant markers were filtered out, and only 1619 markers were suitable for linkage map construction. The 4738 unassigned markers were redistributed in the linkage map, and 111 markers were assigned to known chromosomes in the linkage map. Hence, a total of 1730 markers were used for map construction.

A set of 1589 polymorphic DArTseq markers was mapped to 34 discrete linkage groups, and 141 markers remained unlinked. The linkage map covered 4327.1 cM with an average marker density of one marker per 2.7 cM. Marker density was highest for the B genome, and it covered 1783.1 cM. The A genome map included 1552.2 cM, and the coverage of the D genome was low with 912.5 cM. There was an average marker density of one marker per 2.5 cM in the A genome and one marker per 3.7 cM in the D genome. *YrSen* was also included in the linkage map.

#### 2.2.3. QTL Analysis

Composite interval mapping (CIM) was used to scan the Sentinel/Nyb3 RIL population to detect quantitative trait loci (QTL) associated with stripe rust resistance. CIM analyses detected three consistent QTL on chromosomes 1B, 2A and 3B (Figure 3). All QTL were contributed by Sentinel and were named *QYr.sun-1BL*, *QYr.sun-2AS* and *QYr.sun-3BS*. QTL *QYr.sun-1BL*, *QYr.sun-2AS* and *QYr.sun-3BS* on an average explained 18%, 15.3% and 10.6%, respectively, of the phenotypic variation in adult plant stripe rust response among the Sentinel/Nyb3 RIL population (Table 2).

*QYr.sun-1BL* explained 13–26% of the variation in stripe rust response, peaked at the DArTseq marker 4406454, and was flanked by markers 12766133 and 1074469 (Table 2). Marker 4539050 was the closest to the *QYr.sun-2AS* peak and contributed 11–22% towards phenotypic variation (Table 2). Markers 1095379 and 4440391 defined the QTL interval. *QYr.sun-3BS* corresponded to *YrSen*, and markers 4409093 and 1012045 flanked it (Table 2). This QTL explained 6 to 14% of the phenotypic variation in stripe rust response in the Sentinel/Nyb3 RIL population.

#### 2.2.4. Contribution of Sentinel Alleles

To assess the individual contribution of each QTL towards adult plant stripe rust severity reduction, rust response data collected on a 1–9 scale were converted to percent rust severity (0–100). Comparisons of the mean stripe rust responses of RILs carrying alternative alleles at the peak marker locus were made to confirm the phenotypic contributions of each QTL (Table 3). Detected QTL were considered real when the mean rust response of RILs carrying the positive allele was less than the mean of those possessing the alternate allele. The contributions of peak markers 4406454 (*QYr.sun-1BL*), 4539050 (*QYr.sun-2AS*) and *YrSen* (*QYr.sun-3BS*) are listed in Table 3. *QYr.sun-1BL*, *QYr.sun-2AS* and *QYr.sun-3BS* reduced stripe rust severities by 23–43%, 27–33% and 25–51%, respectively (Table 3).

#### 2.2.5. Mendelization of *QYr.sun-2AS*

Based on the presence of the *QYr.sun-2AS* peak marker 4539050, moderately susceptible response (score 6), and the absence of peak markers for *QYr.sun-1BL* and *QYr.sun-3BS*, Sentinel/Nyb3 RIL#70 (SN#70) was selected as a carrier of *QYr.sun-2AS* singly. SN#70 was crossed with Nyb3 and was advanced to the F_5_ generation through the single seed descent method. F_5_ single plant progenies formed the F_6_ RIL population. The entire RIL population (108 RILs) was phenotyped for stripe rust response variation in the field during the 2017 and 2018 crop seasons. SN#70 was scored ‘6’, and Nyb3 was fully susceptible with a score of ‘9’. Stripe rust responses among the RILs varied from ‘6’ to ‘9’. Monogenic segregation (61 HR:47 HS; χ^2^_1:1_ = 1.81, non-significant at *p* = 0.05 and 1 *d.f*.) for adult plant stripe rust response was observed among the SN#70/Nyb3 RIL population. Based on these results, *QYr.sun-2AS* was temporarily named *YrSen2*.

#### 2.2.6. Interaction of QTL

A comparison of the mean percent stripe rust severities of individual QTL and different QTL combinations across sites and years is presented in Table 4. The critical difference (CD) was calculated for each experiment to see whether the stripe rust severities displayed on RILs with different QTL combinations were significantly different. RILs carrying single QTL exhibited significantly higher stripe rust severities compared to RILs with combinations of two or three QTL. In the 2014 crop season at the LDN site, RILs carrying *QYr.sun-3BS* exhibited significantly lower stripe rust severity (27%) compared to *QYr.sun-2AS* (44%), whereas *QYr.sun-2AS* and *QYr.sun-1BL* did not differ significantly (Table 4). The stripe rust severities of RILs with single QTL did not differ significantly in two other experiments. The RILs carrying *QYr.sun-1BL* + *QYr.sun-2AS* (11.7%) and *QYr.sun-1BL* + *QYr.sun-3BS* (17.1%) produced significantly lower stripe rust severities in 2014 than that exhibited by *QYr.sun-2AS* + *QYr.sun-3BS*-possessing RILs (28.5%). In the remaining two experiments, the severities exhibited by the three different dual combinations of QTL did not differ significantly. The mean rust severity of RILs carrying the Sentinel alleles for all three QTL produced significantly lower stripe rust severities than single- and dual-QTL-carrying RILs (Table 4).

## 3. Discussion

The common wheat cultivar Sentinel has displayed a high level of resistance to stripe rust since its release in 2005. The continuous distribution of stripe rust response variation among the Sentinel/Nyb3 RIL population, when tested against the *Pst* pathotype 134 E16A+ *Yr17*+*Yr27+*, suggested quantitative inheritance of resistance. The involvement of two to three genes in conditioning APR to stripe rust in Sentinel was estimated. This estimation was confirmed with the detection of three consistent QTL, one each on chromosomes 1B, 2A and 3B of Sentinel.

The *QYr.sun-1BL* peaked at the marker 4406454 and mapped in the same genomic region as the APR gene *Yr29*. The presence of *Yr29* in Sentinel was confirmed using a closely linked SNP marker, *Lr46_SNP1G22*. The location of APR gene *Yr29* on the long arm of chromosome 1B has been reported in a number of studies [19]. In the present study, *QYr.sun-1BL* explained 13–26% of the stripe rust variation at an LOD score range of 6.2–10.5. The wide range of LOD scores (2.8–23) and phenotypic variation explanations (4.5–65%) attributed to the *Yr29* locus in several studies have been documented in a review by Rosewarne et al. [19]. The *QYr.sun-1BL* from this study was concluded to be *Yr29*.

The second QTL, *QYr.sun-2AS*, peaked at the marker 4539050 located on the short arm of chromosome 2A. Studies by Agenbag et al. [21], Bansal et al. [20], Bariana et al. [10], Boukhatem et al. [22], Chhuneja et al. [23], Dedryver et al. [24], Hao et al. [25] and Mallard et al. [26] reported QTL in chromosome 2A. Most of these QTL mapped in 2AL based on the location of associated markers, except for markers associated with the stripe rust resistance QTL reported by Bansal et al. [20], which mapped on chromosome 2AS in a durum wheat mapping population. The QTL reported by Bansal et al. [20] was later shown to express resistance at the four-leaf stage and was formally designated *Yr56* (https://wheat.pw.usda.gov/GG3/wgc; accessed on 25 October 2023). *QYr.sun-2AS* from this study does not express resistance at the four-leaf stage and, hence, is likely to be different from *Yr56* and represent a new locus.

The third QTL, *QYr.sun-3BS*, detected in this study peaked at the stripe rust resistance locus, *YrSen*, which was characterized at the four-leaf stage at 21 ± 2 °C and was flanked by DArTseq markers 4409093 and 1012045. Several studies have reported QTL on the short arm of chromosome 3B [19]. The QTL reported in these studies corresponded to the APR gene *Yr30* on chromosome 3BS. The APR gene *Yr30* does not express stripe rust resistance at the four-leaf stage at 21 ± 2 °C. These results indicated that *Yr30* and *YrSen* are different.

The short arm of chromosome 3B also carries the all-stage stripe rust resistance gene *Yr57*. This gene is located at the distal end of chromosome 3BS and is flanked by markers *gwm389* at 2.0 cM proximally and *BS00062676* at a genetic distance of 2.3 cM distally [27]. The post-seedling expression of *QYr.sun-3BS* differentiates it from *Yr57*, which produces near-immune responses at the 2-leaf stage. The stripe rust resistance gene *Yr58* was also mapped on the distal end of chromosome 3BS [28]. *Yr58* can be detected at the 4-leaf stage at 21 ± 2 °C. The detection of *Yr58* and *YrSen* under similar conditions in the greenhouse suggested that these genes may represent the same locus. However, *Yr58* (*QYr.sun-3BS*) explained 34 to 59% of the variation in stripe rust response in a W195/BT-Schomburgk RIL population and, in contrast, *QYr.sun-3BS* (*YrSen*) contributed 6 to 12% to phenotypic variation. In addition, *Yr58* is ineffective against pre-2002 *Pst* pathotypes [28]. Based on these facts, *QYr.sun-3BS* (*YrSen*) is likely to be different from *Yr58*. All these results demonstrated the uniqueness of *QYr.sun-3BS* (*YrSen*) and, hence, it deserves a permanent gene symbol. The expression of *QYr.sun-3BS* at the 4-leaf stage highlights the importance of this locus in the early-growth-stage management of stripe rust.

The combination of all three QTL (*QYr.sun-1BL*, *QYr.sun-2AS* and *QYr.sun-3BS*) detected in Sentinel/Nyb3 RILs exhibited significantly lower stripe rust severities across sites and years (Table 4) compared to RILs with two or single QTL. Likewise, the dual-QTL-carrying RILs displayed stripe rust severities significantly lower than the single-QTL RILs. Such results have been reported in earlier studies [29]. The maintenance of a high level of adult plant stripe rust resistance by Sentinel since 2005 demonstrated the durable nature of *QYr.sun-1BL* (*Yr29*), *QYr.sun-2AS* (*YrSen2*) and *QYr.sun-3BS* (*YrSen*). 

## 4. Materials and Methods

### 4.1. Materials

#### 4.1.1. Plant Material

Sentinel was crossed with a stripe-rust-susceptible genotype, Nyabing 3 (Nyb3), and a mapping population of 117 recombinant inbred line (RIL) F_6_ individuals was developed. Susceptible genotypes Morocco and Nyb3 were included as stripe-rust-susceptible spreaders during field testing.

#### 4.1.2. Pathogen Materials

*Pst* pathotype 134 E16A+17+27+ was used for creating stripe rust epidemics in the field and for greenhouse studies.

### 4.2. Stripe Rust Assessments

#### 4.2.1. Field Assessment

The Sentinel/Nyb3 RIL population was grown in the field as 10 seed hill plots at the Plant Breeding Institute (PBI), Cobbitty, in the 2014 and 2015 crop seasons. The parents Sentinel and Nyb3 were included as controls. Each block of 70 hill plots was surrounded by a mixture of susceptible genotypes (Morocco and Nyb3) to create stripe rust epidemics. The RIL population was evaluated at one experimental site (Lansdowne = LDN) during the 2014 crop season and at two sites (Karalee = K and LDN) in the 2015 crop season. The 2014 experiment was sown in the first week of July, whereas the 2015 experiments were planted in the first week of June at the K site and the second week of June at the LDN site. The experimental area was irrigated to create favorable conditions for stripe rust development. The experiments were artificially inoculated using *Pst* pathotype 134 E16A+17+27+ suspended in light mineral oil (Isopar L^®^) and misted over the whole experiment using an ultra-low-volume Micron sprayer. Further inoculation was done by dropping rusty potted seedlings between susceptible spreader rows. Adult plant stripe rust response assessments were performed using a 1–9 scale (1 = very resistant (VR), 2 = resistant (R), 3 = resistant to moderately resistant (RMR) 4 = moderately resistant (MR), 5 = moderately resistant to moderately susceptible (MRMS), 6 = moderately susceptible (MS), 7 = moderately susceptible to susceptible (MSS), 8 = susceptible (S) and 9 = very susceptible) described in Bariana et al. [30]. The 1–9 scale-based scores were converted into percent severities for comparison of genotypes carrying different numbers of QTL [30].

#### 4.2.2. Greenhouse Assessment

The Sentinel/Nyb3 RIL population and both parents were tested at the two-leaf, three-leaf and four-leaf stages with pathotype 134 E16A+17+r27+ under two temperatures regimes (18 ± 2 °C and 21 ± 2 °C) to observe the expression of resistance under greenhouse conditions. The sowing and inoculation details of the greenhouse experiments are described in Randhawa et al. [31]. Stripe rust assessments were made using a 0–4 scale described in McIntosh et al. [32].

### 4.3. Molecular Mapping

#### 4.3.1. DNA Extraction

Leaf tissues of about 2.5 cm length, picked from at least eight seedlings from each of the RIL and parents, were put in well-labelled 2 mL Eppendorf tubes and dried on silica gel for 3 days. DNA was isolated using the modified CTAB method described in Bansal et al. [20]. DNA samples were quantified using a Nanodrop ND-1000 spectrophotometer (Thermo Fisher Scientific, Wilmington, NC, USA), and 30 ng/µL genomic DNA dilutions were made.

#### 4.3.2. Linkage Map Construction and QTL Analysis

The DNA samples of 92 Sentinel/Nyb3 RILs and the two parents were sent to Diversity Arrays Technology, Canberra, Australia, for DArTseq genotyping. A total of 16,815 DArTseq markers were used to genotype the Sentinel/Nyb3 RIL population and scored ‘1’ for presence and ‘0’ for absence of the target marker. Chi-squared (χ^2^) analysis was performed to check segregation distortion of markers from the expected ratio of 1:1, and markers with Chi-squared values greater than 4 were excluded from the linkage map. Monomorphic markers, redundant markers and markers with more than 10% missing data were excluded from the linkage map. Sentinel/Nyb3 linkage groupings were created using ‘MapDisto’ software version 1.7.5 for MS Windows 2007, and linkage groups were selected based on LOD score of 3. MapManager QTXb20 [33] was used for construction of the linkage map. Composite interval mapping (CIM) was performed using window QTL Cartographer V2.5_011.

### 4.4. Marker Screening for APR Genes Yr18 and Yr29

*Yr18*-linked marker *csLV34* [34] and *Yr18* gene-specific markers *cssfr1*, *cssfr2*, *cssfr3*, *cssfr4*, *cssfr5* and *cssfr6* [35] were used to detect the presence of *Yr18* in parents Sentinel and Nyb3. *Lr34*-carrying cultivar Janz was included as a positive control. PCR amplifications were performed in 10 µL reaction volumes containing 30 ng of genomic DNA, 0.2 mM dNTPs, 1× PCR buffer containing MgCl_2_ (Bioline), 0.5 µM of each forward primer and reverse primer and 0.02 U Immolase Taq DNA polymerase (Bioline (Sydney, Australia) Pty Ltd.). PCR products were separated in 2% agarose gel.

*Yr29*-linked SNP marker *Lr46_SNP1G22* was employed to detect the presence of this gene in the parents. Lalbahadur + *Lr46* and Lalbahadur were included as positive and negative controls, respectively. PCR reactions were performed in 8 µL reaction volumes containing 3 µL of 30 ng/µL genomic DNA, 4 µL of 2× KASP mix (KBioscience LLC, London UK), 0.11 µL of assay mix (containing 12 µM of each allele-specific forward primer and 30 µM of reverse primer) and 0.89 µL of autoclaved double-distilled water. The PCR conditions used included 94 °C for 15 min; 10 touchdown cycles of 94 °C for 20 s and 65–57 °C for 60 s (dropping 0.8 °C per cycle); and 26–35 cycles of 94 °C for 20 s and 57 °C for 60 s. Readings were taken by fluorescence detection of the reactions at 40 °C for 30 s, and the data were analyzed using CFX Manager 3.1 software (Biorad, Hercules, CA, USA).

### 4.5. Statistical Analysis

The minimum number of genes segregating for stripe rust resistance among the Sentinel/Nyb3 RIL population in the field was estimated using Wright’s method [36] adjusted for the level of inbreeding in the original formula [37]. The equation used is as below:n = (GR) 2/4.27 × σ^2^g
where n = minimum number of effective genes, GR = genotypic range and σ^2^g = genetic variance of RILs.

Chi-squared (χ^2^) analysis was performed to determine the goodness of fit of the observed segregation for stripe rust response at the 4-leaf stage with expected genetic ratios and to check for segregation distortion of markers. Critical differences (CDs) were calculated using the following formula: CD = Standard error × t value at *p =* 0.05.

## 5. Conclusions

The combination of three consistent QTL (*QYr.sun-1BL*, *QYr.sun-2AS* and *QYr.sun-3BS*) were demonstrated to condition high level of adult plant stripe rust resistance in wheat cultivar Sentinel. The additive nature of these QTL in conditioning a high level of stripe rust resistance was demonstrated through comparison of recombinant inbred lines (RILs) carrying these QTL in all different combinations. While *QYr.sun-1BL* corresponded to APR gene *Yr29*, *QYr.sun-3BS* and *QYr.sun-2AS* were temporarily named *YrSen* and *YrSen2*, respectively. Based on their uniqueness, *YrSen* and *YrSen2* deserved to be permanently designated. These loci are a valuable addition to a small number of permanently named APR genes. The durability of adult plant stripe rust resistance of Sentinel makes it an excellent donor source of stripe rust improvement in wheat breeding programs.

## Figures and Tables

**Figure 1 plants-13-00129-f001:**
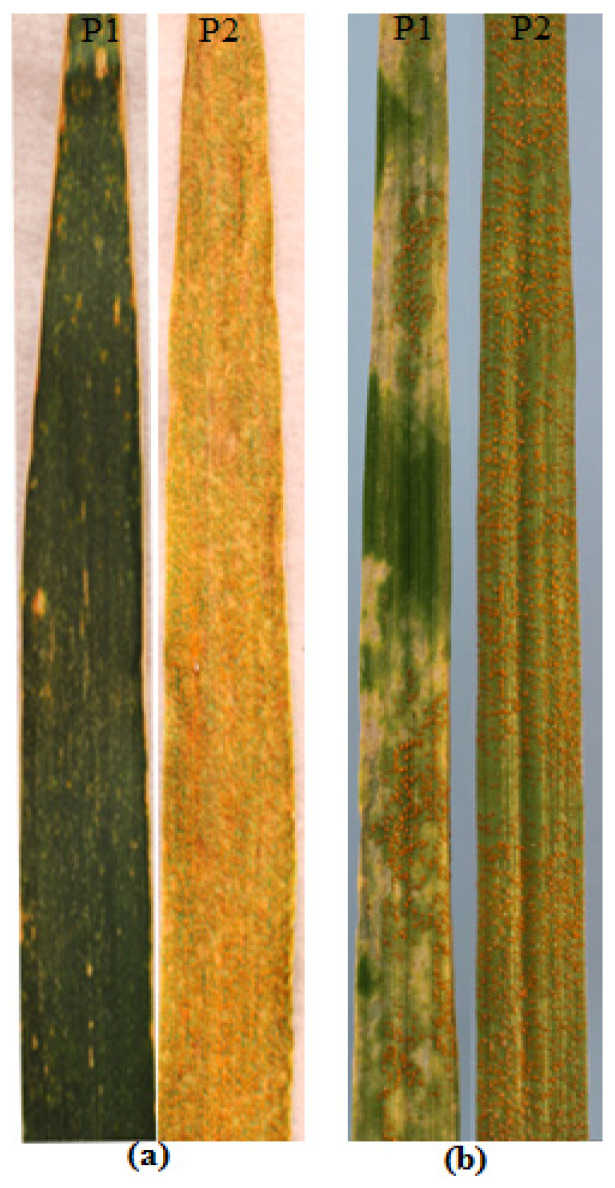
Stripe rust responses of P1-Sentinel and P2-Nyb3 (**a**) in the field and (**b**) at the four-leaf stage in the greenhouse.

**Figure 2 plants-13-00129-f002:**
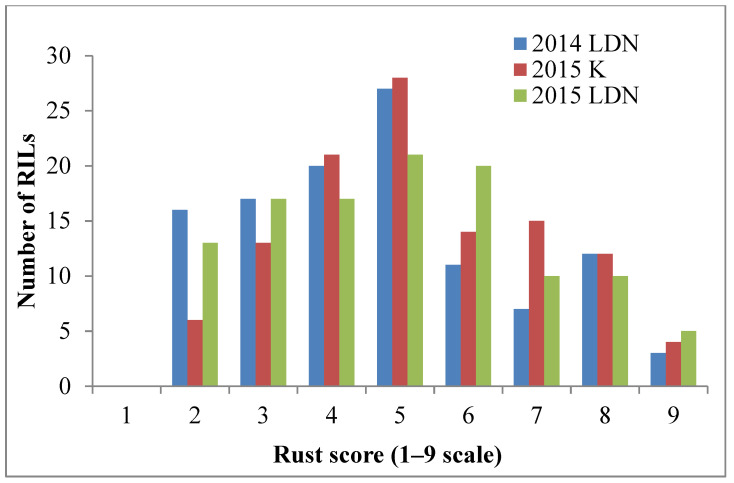
Frequency distribution of FigureSentinel/Nyb3 RILs with respect to adult plant stripe rust response variation (LDN—Lansdowne, K—Karalee).

**Figure 3 plants-13-00129-f003:**
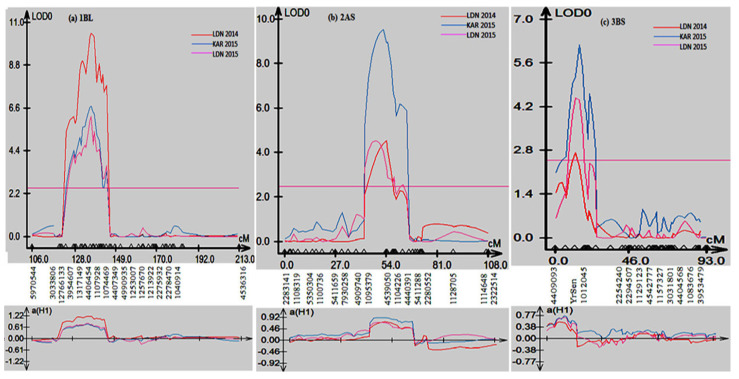
Stripe rust QTL detected on chromosomes (**a**) 1BL, (**b**) 2AS and (**c**) 3BS of Sentinel.

**Table 1 plants-13-00129-t001:** Frequency distribution of Sentinel/Nyb3 F_6_ RILs when tested against the *Pst* pathotype 134 E16A+17+27+ at the four-leaf stage at 21 ± 2 °C.

	Number of RILs		
Response Category	Observed	Expected	χ^2^_1:1_
Homozygous Resistant (HR)	58	58.5	0.004
Homozygous Susceptible (HS)	59	58.5	0.004
Total	117	117	0.008 *

* Table value of χ^2^ at *p* = 0.05 and 1 *d.f.* = 3.84.

**Table 2 plants-13-00129-t002:** Stripe rust QTL detected in the Sentinel/Nyb3 RIL population.

QTL	Season/Site	Peak Marker	Flanking Markers	LOD	*R* ^2^
*QYr.sun-1BL*	2014 LDN	4406454	12766133 & 1074469	10.45	26
	2015 K	4406454	12766133 &1074469	6.69	13
	2015 LDN	4406454	12766133 &1074469	6.18	15
	Mean				18
*QYr.sun-2AS*	2014 LDN	4539050	1095379 & 4440391	4.54	11.4
	2015 K	4539050	1095379 & 4440391	9.54	22
	2015 LDN	4539050	1095379 & 4440391	4.54	13
	Mean				15.3
*QYr.sun-3BS*	2014 LDN	*YrSen*	4409093 & 1012045	2.73	6
	2015 K	*YrSen*	4409093 & 1012045	6.17	13.6
	2015 LDN	*YrSen*	4409093 &1012045	4.47	12.1
	Mean				10.6

**Table 3 plants-13-00129-t003:** Comparison of mean stripe rust severities of RILs carrying Sentinel and Nyb 3 alleles for each QTL.

				Severity (%)		Severity Reduction (%)
QTL	Experiment	Chromosome	Marker	Sentinel Allele	Nyb3 Allele	
*QYr.sun-1BL*	2014 LDN	1BL	4406454	34.6	77.2	42.6
	2015 K	1BL	4406454	56.1	79.4	23.4
	2015 LDN	1BL	4406454	45.0	75.3	30.3
*QYr.sun-2AS*	2014 LDN	2AS	4539050	44.1	77.2	33.1
	2015 K	2AS	4539050	48.0	79.4	31.4
	2015 LDN	2AS	4539050	47.9	75.3	27.4
*QYr.sun-3BS*	2014 LDN	3BS	*YrSen*	26.7	77.2	50.6
	2015 K	3BS	*YrSen*	49.2	79.4	30.3
	2015 LDN	3BS	*YrSen*	50.0	75.3	25.3

**Table 4 plants-13-00129-t004:** Mean stripe rust severities of Sentinel/Nyb3 RILs carrying different QTL combinations.

Loci	Rust Severity (%)		
	2014	2015	
	LDN	K	LDN
*QYr.sun-1BL*	34.58	56.07	45.00
*QYr.sun-2AS*	44.11	48.04	47.91
*QYr.sun-3BS*	26.67	49.17	50.00
*QYr.sun-1BL* + *QYr.sun-2AS*	11.67	34.50	28.75
*QYr.sun-1BL* + *QYr.sun-3BS*	17.06	34.38	30.74
*QYr.sun-2AS* + *QYr.sun-3BS*	28.50	28.75	26.25
*QYr.sun-1BL* + *QYr.sun-2AS + QYr.sun-3BS*	6.72	12.94	13.53
Critical Difference (CD)	9.68 ± 4.98	10.77 ± 5.54	9.88 ± 5.08

## Data Availability

All data are presented in this manuscript.

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
