# Peer review of "A Combination of Three Genomic Regions Conditions High Level of Adult Plant Stripe Rust Resistance in Australian Wheat Cultivar Sentinel"

_plants, 2024, doi:10.3390/plants13010129_

Round 1
Reviewer 1 Report
Comments and Suggestions for Authors
In this manuscript, three consistent QTL QYr.sun-1BL, QYr.sun-2AS, and QYr.sun-3BS for stripe rust resistance were discovered in the wheat cultivar Sentinel and two of them are new resistance loci. The results in this manuscript are useful for the stripe rust resistance breeding program of wheat. This manuscript needs some minor revision as following:
1. The represent figure about the results of detection of Yr18 and Yr19 should be shown.
2. The authors indicated that the resistance loci on 2A and 3B are different from the reported ones, and this should be confirmed using markers.
3. The pedigree of Sentinel and the possible origin of the resistance genes should be described.
This manuscript can be accept after minor revision.
Reviewer 2 Report
Comments and Suggestions for Authors
See the attached file

Needs modifications
Reviewer 3 Report
Comments and Suggestions for Authors
The manuscript described a significant contribution to understanding adult plant resistance to stripe rust in bread wheat. Although several minor changes are suggested, publication of the manuscript is recommended. Since new gene symbols are suggested, the justification for them needs to be clear and concise.
Review notes on manuscript plants-2711218, “Combination of Three Genomic Regions Conditioned High Level of Adult Plant Stripe Rust …”
Line 3: Capitalize “Rust”
Line 24: To reduce possible confusion, using the locus name YrSen1 might be better than YrSen. Also, YrSen1 might have an advantage when scanning articles or literature.
Line 24: The gene symbol YrSen1 should be identified at a suggested temporary symbol because allelism tests with Yr56 were not conducted.
Line 25: The gene symbol YrSen2 should be identified at a suggested temporary symbol because no allelism tests with Yr30 were conducted.
Line 65: Place a semicolon before “however.”
Line 88: Pst should be italics.
Table 1: Check for the correct capitalization of headings in Table 1.
Line 164: Delete the word genome.
Table 2: The title and footnotes for Table 2 are incomplete. Indicate these results are for three field locations. The footnote should provide the full name for each location. Presenting means should be justified statistically. What are the significant digits for the R2 values?
Figure 3: Modify the title to include location information. The QTL were detected in the Sentinel/Nyb3 population with Sentinel contributing the positive allelenly .
Tables 3 and 4: Add a footnote providing location information.
Table 4: How many RILs had each combination of APR genes? Did any of the RIL lack all three APR genes? For the means given in Table 4, are all four digits significant?
Line 230: What makes Sentinel common?
Lines 253-257: No change is necessary. However, genetic background and its interaction with temperature cannot be ruled out as a cause of the differential 4-leaf response.
Lines 256-257: Since no allelism tests with Yr56 were conducted, caution must be included in this statement about being a new locus. The differential 4-leaf responses do not provide any information about the possibility of allelism. This is important because recombinants between Yr56 and YrSen1 might be a means of improving APR resistance to stripe rust in breeding lines.
Line 260: The symbol YrSen or YrSen1 must be identified as a suggested temporary gene symbol because an allelism test with Yr30 was not conducted.
Lines 263-264: Since genetic background effects cannot be ruled out, add “likely are” or “might be” to the statement. Indicate that allelism tests or gene sequences are necessary to confirm that two different loci are involved. If two loci exist, recombination is possible and the level of APR resistance can be enhanced.
Lines 292-294: Provide pedigrees and collection numbers for Sentinel, Nyabing 3, and Morocco. Was Sentinel the male or female parent in the cross? Does the pedigree of Sentinel provide any clues regarding the origin of the APR genes? If Sentinel is a commonly grown cultivar, provide citations indicating distribution and use.
Lines 297 and 310: Pst should be italics.
Lines 360-363: The temperatures should be recorded without a space before °C.
Line 385: The gene symbols YrSen1 and YrSen2 should be identified as suggested temporary symbols because no other authority has approved their use.
Line 485: Capitalization of the article title is incorrect.
Round 2
Reviewer 2 Report
Comments and Suggestions for Authors
It seems ok.
Author Response
Thank you very much.